# Caffeinated beverages intake and risk of deep vein thrombosis: A Mendelian randomization study

**Tong Lin** ⓘ *, **Haiyan Mao**, **Yuhong Jin**

Department of Critical Care Medicine, Ningbo Medical Center Lihuili Hospital Ningbo, Zhejiang, The People's Republic of China

* kai893@foxmail.com

## Abstract

This study aimed to explore the potential link between coffee and tea consumption and the risk of deep vein thrombosis (DVT) through Mendelian randomization (MR) analysis. Employing the MR, we identified 33 single nucleotide polymorphisms (SNPs) as instrumental variables (IVs) for coffee intake and 38 SNPs for tea intake. The investigation employed the inverse-variance weighted (IVW) method to evaluate the causal impact of beverage consumption on DVT risk. Additionally, MR-Egger and MR-PRESSO tests were conducted to assess pleiotropy, while Cochran's Q test gauged heterogeneity. Robustness analysis was performed through a leave-one-out approach. The MR analysis uncovered a significant association between coffee intake and an increased risk of DVT (odds ratio [OR] 1.008, 95% confidence interval [CI] = 1.001–1.015, P = 0.025). Conversely, no substantial causal effect of tea consumption on DVT was observed (OR 1.001, 95% CI = 0.995–1.007, P = 0.735). Importantly, no significant levels of heterogeneity, pleiotropy, or bias were detected in the instrumental variables used. In summary, our findings suggest a modestly heightened risk of DVT associated with coffee intake, while tea consumption did not exhibit a significant impact on DVT risk.

## Introduction

Deep vein thrombosis (DVT) is a pathological condition characterized by the formation of one or more blood clots in the deep veins of the body, typically located in the lower extremities. Although DVT may present with leg pain or swelling, it can also be asymptomatic [1]. More seriously, its potential to cause pulmonary embolism (PE), a life-threatening complication that arises when a clot dislodges and obstructs blood flow within the lungs, underscores its clinical significance [2]. The development of DVT is influenced by lifestyle factors, environmental conditions, and various genetic predispositions, such as sedentary habits, obesity, cancer, and post-surgery effects [3]. While lifestyle factors like sedentary behavior and obesity are strongly linked to DVT, the precise impact of dietary elements, including the consumption of caffeinated beverages, on DVT remains uncertain.

**Data Availability Statement:** All data are linked to the GWAS database. Data for coffee intake was obtained from (https://gwas.mrcieu.ac.uk/datasets/ukb-b-5237/). Data for tea intake was obtained from (https://gwas.mrcieu.ac.uk/datasets/ukb-b-

6066/). The summary data for DVT was obtained from (https://gwas.mrcieu.ac.uk/datasets/ukb-b-12040/).

**Funding:** The author(s) received no specific funding for this work.

**Competing interests:** The authors have declared that no competing interests exist.

Coffee and tea are the two most widely consumed caffeinated beverages globally, both containing biologically active compounds, with caffeine being the most well-known among them. These compounds have been shown to provide various beneficial effects on human health [4,5]. Caffeine has been found to impact the cardiovascular system, causing a rapid increase in heart rate and blood pressure. However, when consumed in moderation, caffeine can potentially reduce the risk of developing cardiovascular disease and may even act as a preventative measure against it [6]. A single study conducted on an older women cohort has explored the link between coffee intake and venous thrombosis in a prospective manner. The findings suggested a slight inverse correlation between coffee consumption and VTE, but after taking into account variables such as body mass index (BMI) and diabetes, the association was no longer significant [7]. Additionally, Enga et al. conducted a prospective cohort study that observed a U-shaped relationship between coffee consumption and venous thromboembolism. Specifically, moderate coffee intake was linked to a decreased risk of venous thromboembolism [8]. These two studies suggest a possible inverse association between coffee and DVT, although the evidence remains inconclusive. Unfortunately, no studies have been found to investigate the association between tea consumption and venous thrombosis. To date, the majority of research has focused on investigating the role of coffee or tea in the pathogenesis of arterial thrombosis and cardiovascular disease (CVD) (e.g. myocardial infarction) [9–12]. As a result, our current understanding of the impact of coffee or tea on the risk of DVT remains limited. However, the association between caffeinated beverages and DVT has not been clearly confirmed in observational studies, and confounding and reverse causality are possible.

Mendelian randomization (MR) is a powerful method that uses genetic variants as instrumental variables to investigate causality between an exposure and an outcome. The use of MR in the study of coffee or tea intake with DVT has several advantages over observational studies. First, genetic variants are randomly allocated at conception, which eliminates the potential for confounding and reverse causality that can occur in observational studies. Second, the effects of genetic variants on coffee or tea intake are not subject to the same bias as self-reported dietary data, reducing measurement error. Finally, MR analysis can provide estimates of causal effects that are less likely to be biased by unmeasured confounding or reverse causality, providing more robust evidence for causal inference. Therefore, the use of MR in investigating the association between coffee or tea intake and DVT risk can provide more reliable and valid results.

## Materials and methods

### Study design

In order to assess the potential causal relationship between caffeinated beverages and DVT, we conducted a two-sample Mendelian randomization analysis [13]. The study adhered to the MR protocol, shown in Fig 1. It rests on three fundamental assumptions underlying the MR study. First, it is assumed that single-nucleotide polymorphisms (SNPs) are highly correlated with the consumption of caffeinated beverages such as coffee or tea. Second, it is assumed that these SNPs are independent of any potential confounding factors that could influence the outcome being studied. Finally, it is assumed that the effects of the SNPs on the development of DVT are solely mediated through the consumption of caffeinated beverages.

### Ethical approval and data sources

This article utilized data from genome-wide meta-analysis (GWAS) [14] that has been ethically reviewed and is publicly available. The summary-level data used in the analysis pertained to traits of interest and were obtained from predominantly European individuals, including both

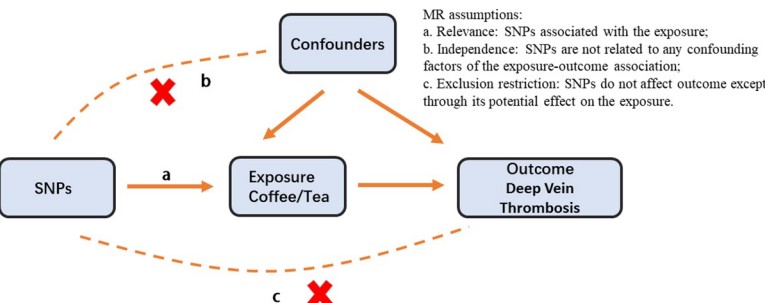

**Fig 1. There are three fundamental assumptions underlying the Mendelian randomization study.**

males and females. Specifically, the study focused on coffee intake (https://gwas.mrcieu.ac.uk/datasets/ukb-b-5237/) and tea intake (https://gwas.mrcieu.ac.uk/datasets/ukb-b-6066/) in a European population consisting of 428,860 and 447,485 study subjects, respectively. The summary data for deep vein thrombosis (DVT) were obtained from the MRC Integrative Epidemiology Unit Consortium (MRC-IEU) in 2018 and were included in UK Biobank (https://gwas.mrcieu.ac.uk/datasets/ukb-b-12040/). This data set comprised a total of 462,933 participants, including 9,241 DVT patients and 453,692 controls.

## Genetic instrumental variables

In our study, we utilized a set of quality control criteria based on the GWAS summary coffee and tea data to select eligible genetic instrumental variables (IVs). Firstly, we employed independent genetic variants that exhibited significant associations with each exposure ($p < 5 \times 10^{-8}$) for each instrument. Subsequently, we carried out the clumping procedure using a window size >10,000 kb and $R^2 < 0.001$ to eliminate linkage disequilibrium (LD). Secondly, we removed SNPs with a minor allele frequency (MAF) of less than 0.01. Thirdly, to mitigate potential pleiotropic effects, we relied on Phenoscanner (https://www.phenoscanner.medschl.cam.ac.uk), a database housing genotype-phenotype associations, to validate the integrity of the chosen instrumental variables [15]. Established risk factors for DVT, including obesity, cancer, and a history of venous thromboembolism, are well-supported in current literature [3]. Utilizing the Phenoscanner platform, we systematically excluded SNPs associated with these known risk factors, thereby reducing the likelihood of confounding influences. Lastly, we calculated the F-statistic (excluded SNPS with F<10), since the included IVs were susceptible to weak IVs [16].

## Statistical analysis

To investigate the potential causal association between Coffee intake, tea intake and the risk of DVT, we used inverse variance weighted (IVW) [17] analysis as the primary method for Mendelian randomization analysis, complemented by weighted-median method [18], MR-Egger method [19], weighted mode and simple mode. Subsequently, Cocrane's Q test and MR-Egger regression were used to assess the presence of heterogeneity and pleiotropy among SNPs, respectively. Furthermore, to test for outlier SNPs, we used MR-PRESSO and performed "leave-one-out" analyses, excluding one SNP at a time to assess the stability of our results. If the IVW method yielded a significant result (p < 0.05) and provided that the beta values of the other methods exhibited a consistent direction, we considered it a positive finding, even if other methods were not significant and no pleiotropy or heterogeneity was identified [20,21]. In cases where horizontal pleiotropy was identified but not heterogeneity, we selected the MR-Egger method. If heterogeneity was detected without pleiotropy, we utilized the weighted-

median method or the multiplicative random-effects IVW method. The MR results were reported as odds ratios (OR) with corresponding confidence intervals (CI) and visualized using forest plots and scatter plots. We conducted all analyses using the TwoSampleMR and MRPRESSO packages in R (version 4.2.1).

## Results

Regarding coffee intake as the exposure factor, we excluded four SNPs (rs1421085, rs62064918, rs476828, rs56113850) associated with risk factors for DVT by Phenoscanner, particularly those linked to obesity and cancer. Additionally, we removed one SNP (rs10119174) exhibiting palindromic allele frequencies related to coffee intake. Similarly, in the context of tea intake as the exposure factor, we excluded rs9937354 due to its association with cancer by Phenoscanner, and rs2783129 due to its palindromic allele frequencies related to tea intake and their correlation with risk factors for DVT. After the clumping process, we identified 33 SNPs (S1 Appendix) and 38 SNPs (S2 Appendix) as instrumental variables to investigate the genetic association between coffee and tea intake and the risk of DVT, respectively. The two-sample MR analysis suggested a modest association, demonstrating that genetically predicted coffee intake was marginally associated with a slight increase in the risk of DVT (OR 1.008, 95% CI = 1.001–1.015, P = 0.025). However, the results from the MR analysis showed no significant causal effect of tea intake on DVT (OR 1.001 95% CI = 0.995–1.007, $P$ = 0.735). In term of orientation and magnitude, there was the consistent result observed in the method of the weighted median and MR-Egger (Table 1). This was further illustrated through scatter plots, encompassing both the weighted mode and simple mode analyses (Figs 2 and 3).

To ensure the robustness of our findings, we conducted sensitivity analyses. Firstly, Cochran's Q test indicated no heterogeneity among the IVs for both coffee ($P_{IVW}$ = 0.451, $P_{MR Egger}$ = 0.474, Table 1) and tea ($P_{IVW}$ = 0.193, $P_{MR Egger}$ = 0.163, Table 1). The symmetry of the funnel plot further supported the absence of heterogeneity (Fig 4). Secondly, the MR-Egger regression results suggested no overall horizontal pleiotropy among all IVs for both coffee ($P$ = 0.234, Table 1) and tea ($P$ = 0.964, Table 1). Additionally, the MR-PRESSO global test did not provide evidence of pleiotropy ($P$ > 0.05, Table 1). Finally, the leave-one-out sensitivity analysis, which involved removing one SNP at a time, yielded consistent results (Fig 5). Based on our analysis, there appears to be a noteworthy link between genetically predicted coffee

**Table 1. Mendelian randomization estimates of the associations between caffeinated beverages intake and risk of deep vein thrombosis.**

| Exposure | Methods of MR | Number of SNP | Beta | OR (95% CI) | P | P For heterogeneity test | P For MR-Egger intercept | P For MR-PRESSO (outliers = 0) |
|---|---|---|---|---|---|---|---|---|
| Coffee intake | IVW | 32 | 0.0082 | 1.008 (1.001–1.015) | 0.025 | 0.451 | 0.234 | 0.482 |
| | MR Egger | 32 | 0.0009 | 1.001 (0.987–1.015) | 0.905 | 0.474 | | |
| | Weighted median | 32 | 0.0031 | 1.003 (0.993–1.013) | 0.548 | | | |
| Tea intake | IVW | 37 | 0.0010 | 1.001 (0.995–1.007) | 0.735 | 0.193 | 0.964 | 0.208 |
| | MR Egger | 37 | 0.0008 | 1.001 (0.987–1.015) | 0.917 | 0.163 | | |
| | Weighted median | 37 | 0.0012 | 1.001 (0.993–1.010) | 0.781 | | | |

MR: Mendelian randomization; IVW: Inverse variance weighted, SNP: Single-nucleotide polymorphism; OR: Odds ratio; CI: Confidence interval.

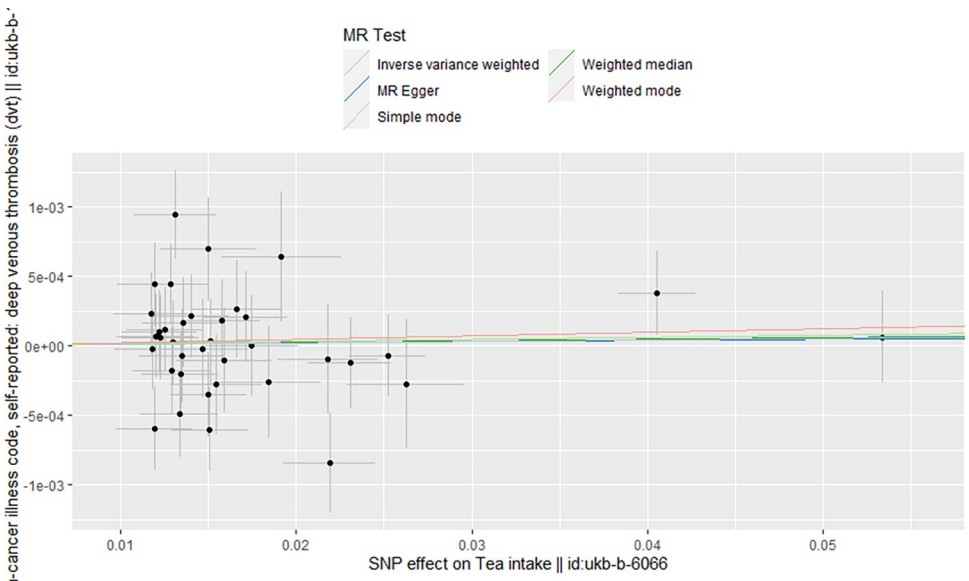

**Fig 2. Scatter plot for the causal effect of coffee intake on DVT risk.** The slope of the straight line indicates the magnitude of the causal association.

intake and a slightly elevated risk of DVT. However, our investigation did not reveal any causal effect between tea intake and this risk. These conclusions were substantiated by multiple sensitivity analyses, underscoring the reliability of our findings.

## Discussion

Deep Vein Thrombosis (DVT) is a significant contributor to cardiovascular disease and it is strongly associated with incidence, mortality and healthcare costs globally. Accurate diagnosis

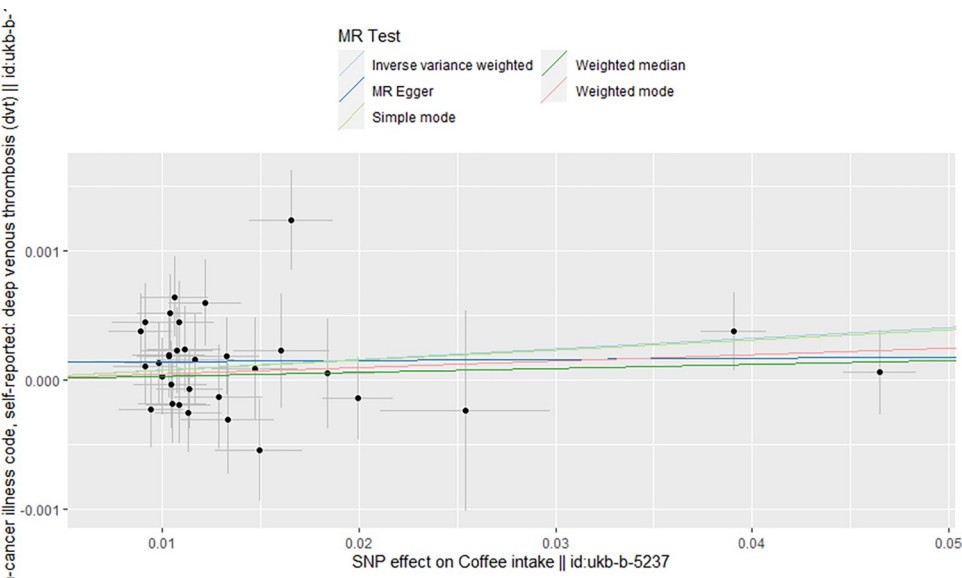

**Fig 3. Scatter plot for the causal effect of tea intake on DVT risk.** The slope of the straight line indicates the magnitude of the causal association.

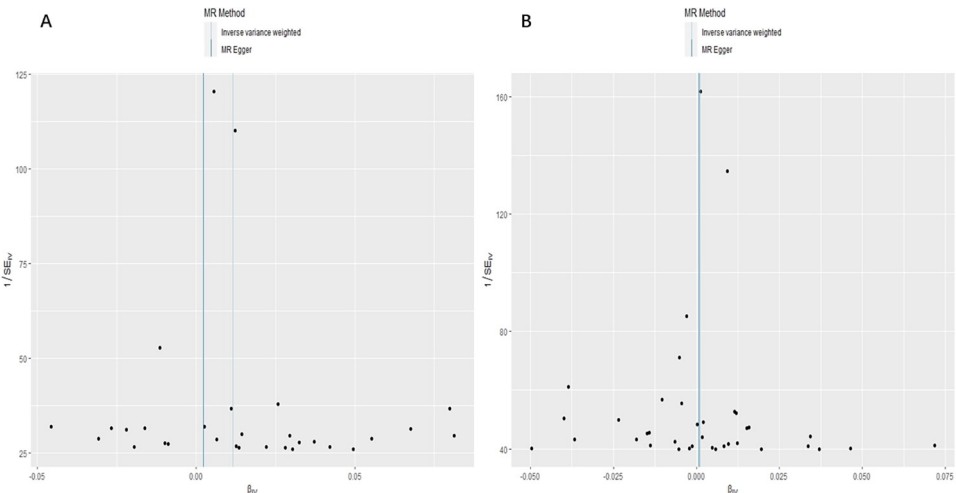

**Fig 4.** Funnel plot for the overall heterogeneity in the effect of coffee intake (A) and tea intake (B) on DVT risk.

and timely elimination of DVT are crucial for reducing the risk of complications and improving patients' quality of life [22]. Therefore, it is imperative to accurately assess the risk factors in DVT patients.

Coffee and tea, the world's top two caffeinated drinks, offer remarkable preventive qualities. Within them lie bioactive dietary polyphenols, presenting a range of valuable therapeutic effects like antioxidant properties, heart health support, neuroprotective abilities, and aid against obesity and high blood pressure [4,5,23]. To date, the majority of research has focused on investigating the role of coffee or tea in the pathogenesis of arterial thrombosis and cardiovascular disease (e.g., myocardial infarction) [24]. However, on the formation of DVT have not been well evaluated. There is currently no conclusive assessment regarding whether coffee and tea have a preventive effect on the formation of DVT. Based on our analysis, there appears to be a noteworthy link between genetically predicted coffee intake and a slightly elevated risk of DVT. Conversely, our results showed no significant association between tea intake and DVT risk in beverages containing caffeine.

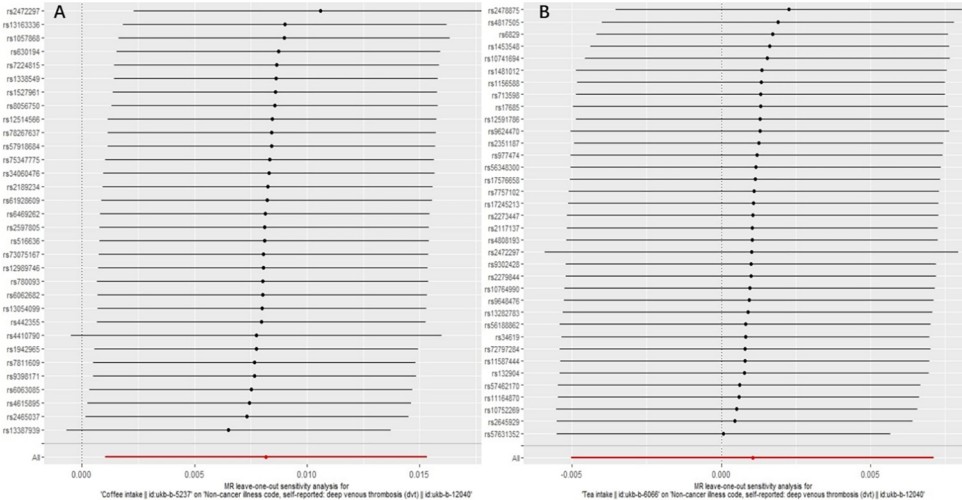

**Fig 5.** Leave-one-out analysis of the effect of coffee intake (A) and tea intake (B) on DVT risk.

Early studies have indicated that certain components in coffee, apart from caffeine, may have the ability to inhibit platelet aggregation, which suggests a potential protective effect against cardiovascular disease [25]. It is widely acknowledged that venous thromboembolism and arterial thrombosis are characterized by distinct underlying mechanisms, locations and treatment modalities. Arterial thrombosis is distinguished by vascular endothelial injury and heightened shear stress, commonly affecting the coronary arteries and cerebrovascular system. Inversely, venous thromboembolism arises from venous stasis and hypercoagulability, typically involving the lower limb veins and pulmonary arteries [26]. Moreover, the effects of coffee on blood coagulation function remain a topic of controversy. A small-scale randomized controlled trial found no significant impact on factor VII levels and fibrinolytic activity after 9 weeks of coffee consumption [27]. In contrast, an earlier experiment demonstrated an immediate increase in fibrinolytic activity upon coffee intake [28]. A recent study discovered reduced levels of von Willebrand factor and factor VIII among coffee drinkers, but found no association with fibrinogen or anticoagulant proteins [29]. These disparate results highlight the complexity of the relationship between coffee and hemostatic factors. Similarly, certain components found in tea, such as polyphenols, catechin, are believed to have anticoagulant and antiplatelet agents properties [30].

However, research has indicated that the consumption of unfiltered coffee has been associated with elevated levels of blood cholesterol and low-density lipoproteins [31]. This effect may be attributed to the presence of diterpenoids in coffee, that have been found to elevate plasma cholesterol levels, leading to increased blood viscosity [32]. Consequently, hyperlipidemia and hyper-cholesterol levels can constitute to a risk of developing conditions such as atherosclerosis and venous thrombosis [33,34]. A meta-analysis reviewed published clinical studies on the relationship between coffee intake and venous thromboembolism (VTE). By including three studies that met the criteria, the results showed that consuming 1–4 cups of coffee per day was associated with an 11% increase in VTE risk, while consuming ≥5 cups of coffee per day was associated with a 25% decrease in risk [35]. Nevertheless, there is insufficient clinical evidence at present to support the preventive effects of coffee on DVT. Research in this area remains limited and results are conflicting.

A Mendelian randomization study showed that tea consumption can reduce the risk of arterial thrombosis [36]. Besides, there seems to be a lack of sufficient research on the relationship between tea consumption and DVT. While this study's findings may not apply to DVT, it suggests that certain components in tea may have antithrombotic effects. Overall, there is currently insufficient evidence to suggest a link between tea consumption and DVT.

In addition, our research is restricted to populations of European ancestry. While this may mitigate bias caused by population stratification, we are still uncertain if the findings can be extrapolated to other populations. Additionally, the presence of different varieties of coffee and tea could potentially impact the research results. Despite these limitations, our study has several advantages. Firstly, this is the first MR study to evaluate the causal relationship between caffeinated beverages like coffee and tea and the risk of DVT. Secondly, this MR study is based on a large sample of GWAS data from European populations, providing us with sufficient power to estimate causal relationships. Thirdly, the study results are unlikely to be influenced by confounding factors.

## Conclusion

This study utilized Mendelian randomization to investigate the direct impact of coffee and tea consumption on the risk of DVT, a condition associated with severe complications like pulmonary embolism. Employing genetic markers as substitutes for beverage consumption, our

findings revealed a marginal elevation in DVT risk with increased coffee intake. Conversely, no substantial effect on DVT risk was observed with tea consumption.

## Supporting information

**S1 Appendix. Results obtained from Mendelian randomization analysis investigating the association between coffee consumption and deep vein thrombosis.**
(XLSX)

**S2 Appendix. Results obtained from Mendelian randomization analysis investigating the association between tea consumption and deep vein thrombosis.**
(XLSX)

## Author Contributions

**Conceptualization:** Tong Lin.

**Data curation:** Tong Lin.

**Formal analysis:** Tong Lin.

**Software:** Haiyan Mao.

**Writing – original draft:** Tong Lin, Yuhong Jin.

**Writing – review & editing:** Haiyan Mao, Yuhong Jin.

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
