## [Decision Letter · Decision Letter 0]

28 Dec 2023

PONE-D-23-30773Caffeinated Beverages Intake and Risk of Deep Vein Thrombosis: A Mendelian Randomization StudyPLOS ONE

Dear Dr. Lin,

Thank you for submitting your manuscript to PLOS ONE. After careful consideration, we feel that it has merit but does not fully meet PLOS ONE’s publication criteria as it currently stands. Therefore, we invite you to submit a revised version of the manuscript that addresses the points raised during the review process.

We look forward to receiving your revised manuscript.

Kind regards,

Eyüp Serhat Çalık

Academic Editor

PLOS ONE

Journal Requirements:

Did you know that depositing data in a repository is associated with up to a 25% citation advantage (https://doi.org/10.1371/journal.pone.0230416)? If you’ve not already done so, consider depositing your raw data in a repository to ensure your work is read, appreciated and cited by the largest possible audience. You’ll also earn an Accessible Data icon on your published paper if you deposit your data in any participating repository (https://plos.org/open-science/open-data/#accessible-data).

**Additional Editor Comments:**

Congratulations to the authors for their valuable study and efforts. Your manuscript has been evaluated by two reviewers and their suggestions are given below. Please respond to these suggestions point by point and make the necessary edits to your manuscript.

In addition, I would like to know; in the conclusion section of your article, you say that higher coffee intake is associated with a slight increase in DVT, but in the results section, can you give a measure of this higher coffee intake? Is there such a measure in genetic variants? Can you specify?

Finally, please have your manuscript professionally edited for language.

Reviewers' comments:

Reviewer's Responses to Questions

**Comments to the Author**

1. Is the manuscript technically sound, and do the data support the conclusions?

Reviewer #1: Partly

Reviewer #2: Partly

2. Has the statistical analysis been performed appropriately and rigorously? 

Reviewer #1: I Don't Know

Reviewer #2: Yes

3. Have the authors made all data underlying the findings in their manuscript fully available?

Reviewer #1: Yes

Reviewer #2: Yes

4. Is the manuscript presented in an intelligible fashion and written in standard English?

Reviewer #1: No

Reviewer #2: Yes

5. Review Comments to the Author

Reviewer #1: First of all, I want to congratulate the group of Lin and co-workers for their efforts! And I want to thank the journal respectively editor for the allowance to review this interesting manuscript. In the following I will go through the following points step by step: relevance, formals / language, material and methods, results, table, figures, conceptualization, conclusion of this review.

Relevance:

For my understanding the author's tackle a topic of relevance.

Formals / Language:

I am not a native speaker; therefore, I usually consult native speakers or language polishing services to have the English checked. I personally feel like, that such a revision of the English would improve the manuscript. For example, at some points there are redundancies, typos, and at least for my feeling potentially misleading phrasings. One example for the phrasing issue can be found in the lines 47 to 49. I am uncertain whether the wording should be "performed U-shaped relationship" or "observed an U-shaped relationship". One example for a typo can be found in line 92, as I think it should be history instead of “History”.

The abbreviation “VTE” is not correctly introduced. The abbreviation is used first in line 45, and introduced in line 48.

Several abbreviations are introduced repeatedly. One example is “IV” for instrumental variables. Once it is introduced in line 54. Second, it is introduced again in line 118. Please, check this.

Material and Methods:

Yet, I don't have experience in using Mendelian Randomization. Thus, I do not think, that I am competent enough to comment on this.

In line 91 and 92 the authors write “currently, there are established risk factors for DVT, such as obesity, cancer, history of venous thromboembolism and so on”. Why do we find this sentence in the material and methods section? There is no further reference to this in this section of the manuscript.

Results:

For my personal feeling very limited results for complete original article (all results are displayed in 24 lines, 1 table, and 2 figures).

Table:

No comment.

Figures:

Figure 1 – How can be assumend, only looking on an single nucleotide polymorphism level, there only is one mediation – knowing all the risk factos pointed out in the manuscript. For my personal understanding of such a complex entity as deep vein thrombosis, such an assumption is at least a questionable simplification.

Conceptualization:

From a general pathological point of view, the analyses performed must be scrutinized.

The authors point out several times, that their study provides information on causality (e.g., lines 10, 15, 59, 197). But, they only assess. Furthermore, they underline that there are known (and for my feeling important) risk factors for deep vein thrombosis such as cancer or obesity. From my perspective, these are potential confounders respectively mediators are only partially addressed on the level of single nucleotide polymorphisms (SNP). Thus, for my understanding, the statement of providing “causality” overestimates the results of the study.

Conclusion of this review:

Taken together I would recommend a rejection and reconceptualization of the study. I truly believe, that there is important information in this approach, but how it is at least reported in this manuscript it is for my feeling not suitable at the moment. The concise and narrowly circumscribed results, for my feeling, would fit something like, for example a “short communication” (or whatever the respective journal names it).

Reviewer #2: 1. The authors of this study investigated the potential causal relationship between coffee and tea intake and the risk of deep vein thrombosis (DVT) using Mendelian randomization (MR) analysis. The research design is appropriate! Nevertheless, I am a bit concerned about the statistical analysis. In particular, the authors reported for the OR =1.001 for the coffee intake and 95% CI = (1.001, 1.015) with p-value .025. The CI is an open interval meaning the endpoints are not part of the CI. Perhaps, this can be explained by the round-off error. Therefore, I would recommend reporting more decimal digits.

2. While the statistical significance can be supported by the p-value, I am wondering if the finding is practically significant.

3. The statements in Lines 141-144 need rephrasing.

4. Lines 288-289: Why did the authors capitalize the title. Furthermore, the year is 1997 not "n.d."

6. PLOS authors have the option to publish the peer review history of their article (what does this mean?). If published, this will include your full peer review and any attached files.

Reviewer #1: No

Reviewer #2: No

---

## [Author Response · Author response to Decision Letter 0]

8 Jan 2024

Response to Reviewers

Dear editor,

I wish to extend my sincerest gratitude to you and reviewers. Your expertise and dedication in guiding our manuscript through the review process have been invaluable. Your insightful comments and constructive feedback have significantly enhanced the quality of our work. Your thorough assessment has not only strengthened the manuscript but also provided us with invaluable perspectives and suggestions.

We deeply appreciate the time and effort you've dedicated to reviewing and improving our manuscript. Your input has been instrumental in refining our research.

Thank you once again for your invaluable support and guidance.

Warm regards,

Tong Lin

Point-by-point reply to editorial comments:

1.In addition, I would like to know; in the conclusion section of your article, you say that higher coffee intake is associated with a slight increase in DVT, but in the results section, can you give a measure of this higher coffee intake? Is there such a measure in genetic variants? Can you specify?

Response: Thank you for your suggestion. Through Mendelian randomization analysis, we found a linear correlation between coffee intake and DVT. Although the linear slope is small (as demonstrated in Figure 2 of the revised article), it suggests a mild increase in DVT risk with increased coffee intake. The ambiguity in our expression led to the editor's question, and we have made the necessary modifications to clarify this in the revised manuscript. At present, we haven't obtained data on the effects of high-concentration coffee intake on genetics, hence limiting further exploration of its relationship with DVT.

2.Finally, please have your manuscript professionally edited for language.

Response: We appreciate your suggestion and understand the importance of ensuring the language in our manuscript meets professional standards. We have taken your feedback into account and will ensure that our manuscript undergoes professional editing to enhance its language quality before the final submission. Thank you for highlighting this aspect, and we are committed to presenting our research in the most clear and polished manner possible.

Point-by-point reply to Reviewers' comments

Reviewers' comments:

Reviewer #1:

1.Formals / Language: I am not a native speaker; therefore, I usually consult native speakers or language polishing services to have the English checked. I personally feel like, that such a revision of the English would improve the manuscript. For example, at some points there are redundancies, typos, and at least for my feeling potentially misleading phrasings. One example for the phrasing issue can be found in the lines 47 to 49. I am uncertain whether the wording should be "performed U-shaped relationship" or "observed an U-shaped relationship". One example for a typo can be found in line 92, as I think it should be history instead of “History”. The abbreviation “VTE” is not correctly introduced. The abbreviation is used first in line 45, and introduced in line 48. Several abbreviations are introduced repeatedly. One example is “IV” for instrumental variables. Once it is introduced in line 54. Second, it is introduced again in line 118. Please, check this.

Response: We truly appreciate your valuable feedback and understanding regarding the importance of linguistic accuracy in scholarly manuscripts. We acknowledge that ensuring a high standard of English is crucial for conveying our research effectively. Your specific examples of potential redundancies, typos, and phrasing concerns are extremely helpful. We will make a concerted effort to address these concerns by revising the manuscript to eliminate redundancies, correct any typographical errors—such as the one highlighted in line 92 ('History' to be changed to 'history')—and modify potentially misleading phrases, marking these revisions in red throughout the text. We have adjusted the wording from 'performed U-shaped relationship' to 'observed a U-shaped relationship' as suggested and have highlighted these modifications in the revised sentences. We have reviewed the manuscript meticulously to ensure that all abbreviations are introduced upon their first appearance and then consistently used throughout the document without repetition.

2.Material and Methods: Yet, I don't have experience in using Mendelian Randomization. Thus, I do not think, that I am competent enough to comment on this. In line 91 and 92 the authors write “currently, there are established risk factors for DVT, such as obesity, cancer, history of venous thromboembolism and so on”. Why do we find this sentence in the material and methods section? There is no further reference to this in this section of the manuscript.

Response: Thank you for your feedback. Why mention DVT-related risk factors such as obesity, cancer, and history of venous thrombosis in the research methodology? This is because the Mendelian randomization study needs to adhere to three fundamental assumptions. First, it is assumed that single-nucleotide polymorphisms (SNPs) are strongly associated with the consumption of caffeinated beverages like coffee or tea. Second, it is assumed that these SNPs are independent of any potential confounding factors that could affect the studied outcome. Finally, it is assumed that the effects of SNPs on the development of DVT are exclusively mediated through the consumption of caffeinated beverages. Obesity, cancer, a history of venous thrombosis, and other risk factors fall under the second assumption as confounding factors. Therefore, in eliminating the SNPs we've selected, we need to exclude those influenced by confounding factors such as obesity, cancer, a history of venous thrombosis, and factors currently regarded as high-risk factors for DVT. That's why we mention them in the research methodology.

3.Results: For my personal feeling very limited results for complete original article (all results are displayed in 24 lines, 1 table, and 2 figures).

Response: Thank you for your feedback. We understand your concern regarding the limited display of results in our original article. While it may seem that our study exhibits a limited scope of outcomes, it aligns with the methodology involving the use of the two-sample Mendelian randomization. Taking your valuable input into account, we have made efforts to address this concern by including two additional figures, Figure 2 and Figure 3, to further elucidate the relationship between coffee and tea consumption in relation to DVT in the revised version.

4.Figures: Figure 1 – How can be assumend, only looking on an single nucleotide polymorphism level, there only is one mediation – knowing all the risk factos pointed out in the manuscript. For my personal understanding of such a complex entity as deep vein thrombosis, such an assumption is at least a questionable simplification.

Response: Thank you for your thoughtful consideration. The assumption we made regarding the mediation through a single-nucleotide polymorphism (SNP) level stems from the application of Mendelian randomization in our study design. While we acknowledge the complexity inherent in deep vein thrombosis (DVT), the Mendelian randomization framework operates under certain assumptions to establish causal relationships. In the context of our study, the assumption of SNP mediation is based on the fundamental principles of Mendelian randomization. This approach allows us to explore causality by leveraging genetic variants as proxies for exposures. However, we understand and appreciate the multifactorial nature of DVT, involving various risk factors beyond the SNP level. Our aim in assuming SNP mediation through caffeinated beverage consumption was to establish a specific pathway as a focus within the broader complexity of DVT etiology. We agree that DVT is a multifaceted condition influenced by numerous factors, and our study's approach, utilizing Mendelian randomization, centers on delineating specific causal relationships at a genetic level within this intricate landscape.

We greatly appreciate the reviewer's suggestions. To provide a more in-depth understanding of the principles behind the two-sample Mendelian randomization study, we have added a section titled 'Study Design' within the methodology. This aims to offer readers a deeper insight into the fundamental principles of this method. Additionally, in response to the reviewer's feedback, we have modified Figure 1 to make it more intuitive and easily comprehensible. These adjustments are intended to ensure that our research methodology and findings are presented clearly to the readers. We are grateful for the valuable guidance provided by the reviewer, which has contributed to the quality of our study.

5.Conceptualization: From a general pathological point of view, the analyses performed must be scrutinized. The authors point out several times, that their study provides information on causality (e.g., lines 10, 15, 59, 197). But, they only assess. Furthermore, they underline that there are known (and for my feeling important) risk factors for deep vein thrombosis such as cancer or obesity. From my perspective, these are potential confounders respectively mediators are only partially addressed on the level of single nucleotide polymorphisms (SNP). Thus, for my understanding, the statement of providing “causality” overestimates the results of the study.

Response: We appreciate your thorough evaluation of our study. Your insights regarding conceptualization and the assessment of causality are duly noted. Our intention was to highlight potential causal pathways between single-nucleotide polymorphisms (SNPs) associated with caffeinated beverage consumption and the risk of deep vein thrombosis (DVT). However, we acknowledge the complexity inherent in inferring causality solely from SNP associations. While our study aimed to elucidate specific genetic associations, we understand the limitations in comprehensively addressing known risk factors like cancer or obesity, which may act as potential confounders or mediators at levels beyond SNPs. Your point about the statement on causality is well-taken, and upon reflection, we recognize the need for caution in our claims. We will modify our assertions in the manuscript to accurately reflect the scope of our findings as associations rather than definitive causal relationships. Your feedback prompts us to refine our interpretations and ensures the accuracy and integrity of our study. Thank you for your valuable input.

Reviewer #2: 

1.The authors of this study investigated the potential causal relationship between coffee and tea intake and the risk of deep vein thrombosis (DVT) using Mendelian randomization (MR) analysis. The research design is appropriate! Nevertheless, I am a bit concerned about the statistical analysis. In particular, the authors reported for the OR =1.001 for the coffee intake and 95% CI = (1.001, 1.015) with p-value .025. The CI is an open interval meaning the endpoints are not part of the CI. Perhaps, this can be explained by the round-off error. Therefore, I would recommend reporting more decimal digits.

Response: Thank you for your careful review and insightful observations regarding our study's statistical analysis. We acknowledge and deeply regret the oversight that led to the error in reporting the odds ratio (OR) for coffee intake. In the original data, the OR obtained using the IVW statistical method was indeed 1.008209, mistakenly rounded to 1.001. We have rectified this error by accurately adjusting the reported OR to 1.008, and we have conducted a thorough reevaluation of all data presented in the table to ensure precision and accuracy. We appreciate your suggestion to report more decimal digits to avoid open intervals in confidence intervals. Your guidance has been invaluable in ensuring the accuracy of our findings. We apologize for any confusion caused and are grateful for the opportunity to correct this oversight.

2.While the statistical significance can be supported by the p-value, I am wondering if the finding is practically significant.

Response: Thank you for your evaluation and concern. The issue of practical significance is indeed crucial. While there have been studies linking coffee intake to DVT, we acknowledge the presence of uncertainties and potential research biases in those studies. Taking this into consideration, the Mendelian randomization study design allows us to better mitigate confounding factors. This method aids in reducing the interference of other variables on the study outcomes, thereby facilitating a clearer assessment of the relationship between coffee intake and DVT. We will further explore these points in depth while discussing practical significance.

3.The statements in Lines 141-144 need rephrasing.

Response: Thank you for your careful review. The sentence has been polished and highlighted in red in the refined rendition: “Coffee and tea, the world's top two caffeinated drinks, offer remarkable preventive qualities. Within them lie bioactive dietary polyphenols, presenting a range of valuable therapeutic effects like antioxidant properties, heart health support, neuroprotective abilities, and aid against obesity and high blood pressure”.

4.Lines 288-289: Why did the authors capitalize the title. Furthermore, the year is 1997 not "n.d."

Response: Thank you for your guidance and feedback. We apologize for the issues with the citation format. We've identified that it was due to the version of the reference manager and have now thoroughly revised the reference format in accordance with the requirements of the journal "PLOS ONE." We've cross-checked the journal's guidelines and reorganized the citations to ensure compliance with the specified requirements.

---

## [Editor Report · Decision Letter 1]

21 Jan 2024

Caffeinated Beverages Intake and Risk of Deep Vein Thrombosis: A Mendelian Randomization Study

PONE-D-23-30773R1

Dear Dr. Lin,

We’re pleased to inform you that your manuscript has been judged scientifically suitable for publication and will be formally accepted for publication once it meets all outstanding technical requirements.

Kind regards,

Eyüp Serhat Çalık

Academic Editor

PLOS ONE
---

## [Editor Report · Acceptance letter]

2 Feb 2024

PONE-D-23-30773R1 

PLOS ONE

Dear Dr. Lin, 

I'm pleased to inform you that your manuscript has been deemed suitable for publication in PLOS ONE. Congratulations! Your manuscript is now being handed over to our production team.

Kind regards, 

on behalf of

Dr. Eyüp Serhat Çalık 

Academic Editor

PLOS ONE